# Zero-Shot Transfer with Deictic Object-Oriented Representation in Reinforcement Learning

**Ofir Marom[1], Benjamin Rosman [1,2]**
[1]University of the Witwatersrand, Johannesburg, South Africa
[2]Council for Scientific and Industrial Research, Pretoria, South Africa

## Abstract

Object-oriented representations in reinforcement learning have shown promise in transfer learning, with previous research introducing a propositional object-oriented framework that has provably efficient learning bounds with respect to sample complexity. However, this framework has limitations in terms of the classes of tasks it can efficiently learn. In this paper we introduce a novel deictic object-oriented framework that has provably efficient learning bounds and can solve a broader range of tasks. Additionally, we show that this framework is capable of zero-shot transfer of transition dynamics across tasks and demonstrate this empirically for the Taxi and Sokoban domains.

## 1 Introduction

A longstanding objective in reinforcement learning (RL) is transfer learning, where the aim is to accelerate learning in an unseen task using knowledge gained in previously learned tasks [14]. Various Markov decision process (MDP) representations have shown promise in this regard [5, 6, 7, 11, 9]. A common choice among such representations is one where components of an MDP are described with objects [7, 4, 12, 8]. In particular, Propositional Object-Oriented MDPs (Propositional OO-MDPs) [4] introduce a framework to represent the state-space of an MDP in terms of objects while the transition dynamics are represented in terms of propositional preconditions over object classes that map to effects over attributes of the object classes. An appealing property of Propositional OO-MDPs is that the transition dynamics have provably efficient learning bounds for deterministic environments that have been shown to outperform competing model-based approaches for certain domains [3].

Unfortunately, the core restriction of Propositional OO-MDPs that the preconditions be described only in terms of propositions tends to be a strong one and, as even the original authors point out, precludes efficient learning of certain classes of tasks [3]. Such tasks include those where it is required to distinguish between different objects of the same object class. As a specific example of this, and further elaborated in subsection 2.3, consider the Sokoban domain where a person attempts to push a box but cannot do so if that box is adjacent to a wall. In this case, there is no way of tying the box that is adjacent to the person with the box that is adjacent to the wall with propositions. To accommodate such tasks, the transition dynamics for Propositional OO-MDPs must be appended with first-order predicates [1]. However, this impacts the learning efficiency as well as the ability to transfer between tasks since adding more objects now increases the number of preconditions that the transition dynamics depend on.

In this paper we propose to overcome this limitation of Propositional OO-MDPs with a novel deictic object-oriented representation, Deictic OO-MDPs. The key insight behind Deictic OO-MDPs is the concept of deictic predicates. Deictic predicates are grounded only with respect to a central

deictic object, therefore that object may relate itself to non-grounded object classes, but not to other grounded objects. Returning to the Sokoban domain, a deictic predicate over boxes allows a specific box to ascertain whether *any* wall is adjacent to it, but not whether a *specific* wall is adjacent to it. As we formalise in section 2, Deictic OO-MDPs are defined in terms of a schema (or template) and so for all tasks instantiated from that schema the number of preconditions that the transition dynamics depend on remains constant. This makes transfer of the transition dynamics possible across all tasks instantiated from that schema, which is illustrated empirically in section 4. Furthermore, Deictic OO-MDPs allow for the efficient learning of transition dynamics for classes of tasks not possible with propositional frameworks, as discussed in section 3.

## 2 Framework

### 2.1 Background

Most commonly an RL task is described as a discrete-time, finite-state and finite-action Markov decision process (MDP) [13]. Given an MDP, $M$, a well-known model-based algorithm to learn a near-optimal policy for $M$ is $R_{max}$ [1], which is known to have polynomial sample complexity. The KWIK (knows what it knows) [10] framework generalises $R_{max}$ to a broader range of representations, such that if the transition dynamics can be learned with polynomial sample complexity under the KWIK protocol, it is possible to construct an $R_{max}$ type algorithm to learn a near-optimal policy for $M$. The main requirement of the KWIK framework is that when an agent is required to make a prediction of the next state distribution given the agent's current state and action, the agent may choose to return $\perp$ instead, meaning that the agent is unable to make an accurate prediction as it has yet to explore the environment sufficiently. Furthermore, the number of times the agent may return $\perp$ must have polynomial bound (which is called the KWIK bound).

The KWIK framework has been used in conjunction with object-oriented representations for efficient learning [4]. The main idea behind object-oriented representations is that the state-space is made up of grounded objects that are instantiations of object classes. Such representations for the state-space were introduced with Relational MDPs that define a domain in terms of a schema [7].

Formally, the state-space for such a schema consists of a set of object classes $\mathfrak{C} = \{C_i\}_{i=1}^{N_{\mathfrak{C}}}$. Each object class $C \in \mathfrak{C}$ has a set of attributes $Att(C) = \{C.\alpha_i\}_{i=1}^{N_C}$ and each attribute $C.\alpha \in Att(C)$ of an object class has a domain $Dom(C.\alpha)$. Given a schema, a grounded state-space is instantiated by first selecting a grounded object set which consists of $n$ objects $O = \{o_i\}_{i=1}^n$ where each $o \in O$ is an instance of some object class $C$. The value of attribute $C.\alpha$ for object $o$ is denoted by $o.\alpha$. Then the grounded state-space, denoted $S_O$, is an assignment of each $o.\alpha$ for all objects in $O$. The schema state-space, denoted $S$, is the set of all states for all possible object sets $O$.

To make the notion of a schema state-space concrete, consider the classical Taxi domain [2] where a taxi in a gridword has the task of picking up a passenger at some pickup location and dropping them off at some destination location. The actions available to the taxi are $North$, $East$, $South$, $West$, $Pickup$ and $Dropoff$ while walls limit the taxi's movements. We introduce a more general extension of this domain called the all-passenger any-destination Taxi domain where a taxi is tasked to pick up multiple passengers and drop each of them off at one of any destination locations. The taxi can only pick up one passenger at a time, so if a passenger is already in the taxi and the $Pickup$ action is taken while the taxi is at the pickup location of another passenger, the state does not change.

We can represent this more general Taxi domain with four object classes: $Taxi$, $Wall$, $Passenger$ and $Destination$. Each object class has attributes $x$ and $y$ for their location on the grid. Object class $Wall$ has an additional attribute $pos$ to mark one of four positions in a square, while $Passenger$ has additional attributes $in\text{-}taxi$ and $at\text{-}destination$ to indicate if the passenger is in a taxi and at a destination respectively. Given the schema for this domain we can instantiate a set of grounded objects from the object classes and a resulting MDP. Figure 1 shows sample states of the schema.

Propositional OO-MDPs and Relational MDPs use similar state-space representations as described above. Where they differ is in the definition of their transition dynamics. While Relational MDPs use aggregation for preconditions [7], Propositional OO-MDPs use propositions for preconditions [4]. Furthermore, the work on Relational MDPs assumes that the transition dynamics are known and the focus is on learning generalised plans. Meanwhile, Propositional OO-MDPs assume that the transition dynamics are unknown and need to be learned through interaction with the environment.

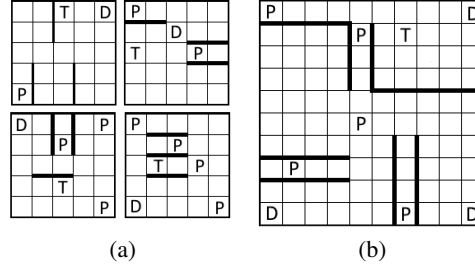

Figure 1: $P$ marks a passenger; $D$ marks a destination; $T$ marks a taxi; thicker lines mark walls. Five possible states from the all-passenger any-destination Taxi domain schema.

## 2.2 Deictic object-oriented representation

Our Deictic OO-MDP framework uses the schema state-space representation described in subsection 2.1 while deictic predicate preconditions are used to define the schema transition dynamics as described in this section. Let $A$ be a set of actions. Then for each attribute $C.\alpha$ and action $a \in A$ define a set of effects $E_{a,C.\alpha} = \{e_i : Dom(C.\alpha) \to Dom(C.\alpha)\}_{i=1}^{K_{a,C.\alpha}}$. Define a set of deictic predicate preconditions $F_{a,C.\alpha} = \{f_i : \mathcal{O}[C] \times S \to \mathcal{B}\}_{i=1}^{D_{a,C.\alpha}}$, where $\mathcal{O}[C]$ is a set that contains objects with all possible attribute value assignments that are instances of $C$, and $\mathcal{B} = \{0, 1\}$. Then the probabilistic transition dynamics for $C.\alpha$ and $a$ are defined by $P_{a,C.\alpha} : \mathcal{B}^{D_{a,C.\alpha}} \times E_{a,C.\alpha} \to [0, 1]$. The schema transition dynamics $P$ is the set of transition dynamics for all attributes and actions, $P = \{P_{a,C.\alpha} | C \in \mathfrak{C}, C.\alpha \in Att(C), a \in A\}$. The schema reward dynamics are defined by $R : S \times A \times S \to \mathbb{R}$.

Given an object set $O$ we can instantiate a grounded MDP $M_{O,\rho} = (S_O, A, P, R, \gamma, \rho)$ where $\gamma$ is a discount rate and $\rho$ is a distribution over initial states. Then if an agent is currently in state $s \in S_O$ and takes action $a$, the transition dynamics for $M_{O,\rho}$ operate as follows: for each object $o$ in $s$ that is an instance of $C$ and each attribute $C.\alpha$ we compute the Boolean truth values $B = \{f_i(o, s)\}_{i=1}^{D_{a,C.\alpha}}$ for the deictic predicates in $F_{a,C.\alpha}$. Then for an effect $e \in E_{a,C.\alpha}$ we compute $P_{a,C.\alpha}(B, e)$ which returns the probability of $e$ occurring given $B$. This implies a distribution over effects which in turn implies a distribution over the attribute values of $o$ by applying the effect to $o.\alpha$ in $s$ and obtaining $e(o.\alpha) = o.\alpha'$ in $s'$.

For example, consider attribute $Taxi.x$ and action $East$ for the all-passenger any-destination Taxi domain introduced in subsection 2.1. We can define a set of relative effects $Rel_i(x) \to x + i$ that produce a shift of $i$ squares from the current location $x$, as well as a deictic predicate $T_E(taxi, s)$ that returns 1 if $taxi$ has a wall one square to its east in $s$, otherwise 0. Then the transition dynamics can be described for any $taxi$ object as $T_E(taxi, s) = 1 \implies taxi.x \leftarrow Rel_0(taxi.x)$ with probability 1 and $T_E(taxi, s) = 0 \implies taxi.x \leftarrow Rel_1(taxi.x)$ with probability 1. For a slightly more complex example consider attribute $Passenger.in\text{-}taxi$ and action $Pickup$. The transition dynamics for this attribute depend on three preconditions for which we require a deictic $passenger$ object: is a taxi on the same square as $passenger$? Is $passenger.at\text{-}destination$ true? Is there any passenger in a taxi?

The key insight with Deictic OO-MDPs is that the parameters we pass to each precondition in $F_{a,C.\alpha}$ are a grounded deictic object $o$ that must be an instance of $C$ and $s$ which is a state *of the schema*, not a grounded state. As a result these preconditions may not refer to specific objects in $s$; however, they may relate $o$ to object classes of the schema. For example, with $T_E(taxi, s)$ as defined above only $taxi$ is grounded while we never refer to a grounded object in $s$.

Given a resulting state $s'$, the reward dynamics operate by computing $R(s, a, s')$. For the purposes of this paper we will assume that $R$ is known while $P$ needs to be learned. A subtle but important requirement for our definition of $P_{a,C.\alpha}$ to be correct is that the set $E_{a,C.\alpha}$ must be *invertible* so that if the current value assignment for some attribute $C.\alpha$ of a grounded object $o$ in state $s$ is $o.\alpha$ and we take action $a$ to subsequently observe $o.\alpha'$ in $s'$ then there must be a unique $e \in E_{a,C.\alpha}$ such that $e(o.\alpha) = o.\alpha'$. Clearly if this requirement is not met then either the effects are not able to correctly capture the true transition dynamics or there are duplicate effects that lead to the same $o.\alpha'$ which creates ambiguity over which effect occurred.

### 2.3 Limitations of propositional object-oriented representation

It is beneficial to contrast the Deictic OO-MDP representation introduced in subsection 2.2 to Propositional OO-MDPs. Propositions alone are insufficient to represent the transition dynamics for the all-passenger any-destination Taxi domain. To see this, suppose we have two passenger objects and the proposition $On(Taxi, Passenger)$ with truth value 1. This can be translated as: a taxi is on the same square as a passenger is true. Clearly this information is insufficient to determine which passenger's $in\text{-}taxi$ attribute should change given the $Pickup$ action. To overcome this we must resort to first-order predicates over grounded objects of the form $On(Taxi, passenger_1)$ and $On(Taxi, passenger_2)$. Note that the number of preconditions changes as we add more passenger objects which complicates both learning and transfer procedures.

As a further example, consider the Sokoban domain where a person is required to push boxes to some storage locations. However, the person cannot push a box if that box is adjacent to another box or a wall. As discussed by the authors of Propositional OO-MDPs such domains are not suitable for propositional representations [3]. Consider the propositions $T_W(Box, Person)$ and $T_E(Box, Wall)$ both with truth value 1. The first term translates as: a box has a person one square to its west is true, while the second translates as: a box has a wall one square to its east is true. Then the conjunction $T_W(Box, Person) = 1 \wedge T_E(Box, Wall) = 1$ is insufficient to determine the transition dynamics of the box's $x$ attribute when taking action $East$ since there is no way to know if the terms are referring to the same box. See Figure 2 for illustration. Deictic OO-MDPs are able to represent these conditions without ambiguity. Consider the $Box.x$ attribute with action $East$. Define $f_1(box, s)$ to return 1 if $box$ has a person one square to its west in $s$, otherwise 0. Define $f_2(box, s)$ to return 1 if $box$ has a wall one squares to its east in $s$, otherwise 0. Then $f_1(box, s) = 1 \wedge f_2(box, s) = 1 \implies box.x \leftarrow box.x + 0$.

In fact, representing the transition dynamics for this domain in terms of Deictic OO-MDPs is straightforward. For a deictic $person$ object there are four preconditions when the action $East$ is taken: is there a box one square east of $person$? Is there a box two squares east of $person$? Is there a wall one square east of $person$? Is there a wall two squares east of $person$? Meanwhile for a deictic $box$ object there are three preconditions: is there a person one square west of $box$? Is there a wall one square east of $box$? Is there a box one square east of $box$? The other actions are analogous.

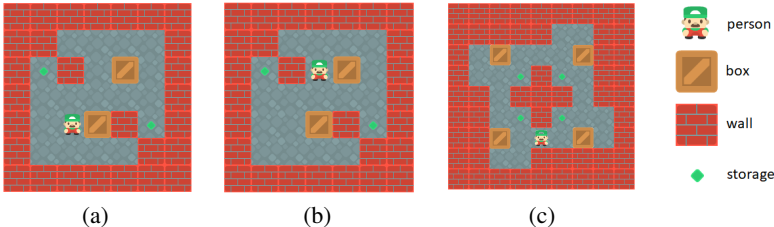

(a)          (b)          (c)

Figure 2: For action $East$ and the box adjacent to the person, in (2a) the effect is $box.x \leftarrow box.x + 0$; in (2b) the effect is $box.x \leftarrow box.x + 1$ while the conjunction $T_W(Box, Person) = 1 \wedge T_E(Box, Wall) = 1$ is true in both cases. (2c) is a more complex Sokoban task with four boxes from the "Micro-Cosmos" level pack which requires 209 steps to solve under an optimal policy.

## 3 Learning transition dynamics

Given a set of $D$ preconditions we want to learn the transition dynamics for each attribute $C.\alpha$ and action $a$. If the transition dynamics are deterministic this can be done using memorisation [10] with KWIK bound $2^D$. However, this is prohibitive if $D$ is large. Propositional OO-MDPs introduce a learning algorithm called $DOORMAX$ [4] for deterministic transition dynamics that, under certain assumptions, has provably efficient KWIK bounds. Moreover, $DOORMAX$ is able to learn from multiple hypothesised effect sets and determine the correct effect set for each attribute and action.

The main assumption required for $DOORMAX$ to be correct is that the transition dynamics for each attribute and action must be representable as a full binary tree with propositions at the non-leaf nodes and effects at the leaf nodes. Furthermore, each possible effect of an effect set can only occur

at most at one leaf node of the tree, except for a special effect called a "failure condition" that may occur at multiple leaf nodes. A failure condition implies that globally no attribute changed when an action was taken i.e. $s = s'$ when $a$ is taken. See Figure 3a for how this represented for the $Taxi.x$ attribute with action $East$.

The intuition behind $DOORMAX$ is that in many cases the number of preconditions that an effect depends on is much smaller than $D$. Furthermore, since an effect can occur at most once in the tree, we can invalidate many terms with a small number of observations by using conjunctions. For example, suppose we observe the following different sets of terms that each produce the same effect $Taxi.x \leftarrow Taxi.x + 1$: $T_1 = \{T_N(Taxi, Wall) = 0, T_E(Taxi, Wall) = 0, T_S(Taxi, Wall) = 0, T_W(Taxi, Wall) = 0\}$ and $T_2 = \{T_N(Taxi, Wall) = 1, T_E(Taxi, Wall) = 0, T_S(Taxi, Wall) = 1, T_W(Taxi, Wall) = 1\}$. Then we can determine from only two observations that the relevant terms for this effect are the ones that occur in $T_1 \wedge T_2$ i.e. $T_E(Taxi, Wall) = 0$. As failure conditions can occur at multiple leaf nodes they need to be learned on a case by case basis.

We adapt the $DOORMAX$ algorithm to Deictic OO-MDPs, which we call $DOORMAX_D$. The main difference for $DOORMAX_D$ is that we remove the notion of a global failure condition because this depends on a grounded state comprised of grounded objects. Meanwhile the transition dynamics of our representation are schema based to allow for transferability across tasks. Instead we require that all effects apply to a single attribute. See Figure 3b for how this is represented for the $Taxi.x$ attribute with action $East$. To achieve this, we introduce a partition function over effects that groups them into those that can occur at most at one leaf node and those that can occur at multiple leaf nodes.

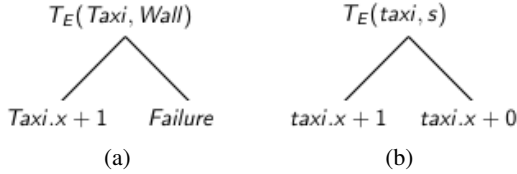

(a)    (b)

Figure 3: Full binary tree structure for the transition dynamics of $Taxi.x$ attribute and action $East$. (3a) for Propositional OO-MDPs; (3b) for Deictic OO-MDPs. Right branches represent a truth value of 1.

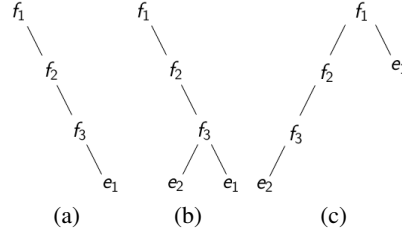

(a)    (b)    (c)

Figure 4: Binary trees induced by $\mathcal{T}_{e_1}$ and $\mathcal{T}_{e_2}$. Right branches represent a truth value of 1.

Let $F$ be a set of deictic predicate preconditions and $E$ be a set of effects.

**Definition 1.** A term is a tuple $(f, b)$ where $f \in F$ and $b \in \mathcal{B}$. A set of terms is denoted by $T$ and a set that contains sets of terms is denoted by $\mathcal{T}$.

**Definition 2.** Term $(f_1, b_1)$ mismatches term $(f_2, b_2)$ if $f_1 = f_2$ and $b_1 \neq b_2$

**Definition 3.** $\Pi : E \rightarrow \mathcal{B}$ is a binary partition function over $E$ and assigns each effect in $E$ to one of two partitions, 0 or 1. We call the tuple $g = (E, \Pi)$ an effect type. Denote by $K_0^g$ and $K_1^g$ the number of effects in partition 0 and 1 respectively. We use . notation to refer to an element in a tuple $g$, so for example $g.E$ refers to $E$ in $g$

**Definition 4.** Let $g$ be an effect type. Let $M > 1$ be a constant. Then $Tree(g, F, M)$ is the set of all full binary trees such that non-leaf nodes are a subset of $F$ and leaf nodes are a subsets of $g.E$. Furthermore, if $g.\Pi$ assigns an effect in $g.E$ to partition 1 then that effect can occur at most at one leaf node and we call that effect conjunctive, otherwise that effect may occur at most at $M$ leaf nodes and we call that effect disjunctive.

**Theorem 1.** For attribute $C.\alpha$ and action $a$ let $\hat{F}$ be a set of size $D$ that contains hypothesised deictic predicate preconditions on which the transition dynamics of $C.\alpha$ when taking action $a$ may depend. Let $\hat{\mathcal{G}} = \{g_i\}_{i=1}^N$ be a set of size $N$ that contains hypothesised effect types where each $e \in g_i.E$ has domain and range $Dom(C.\alpha)$. Let $K_0 = \max_{g \in \hat{\mathcal{G}}} K_0^g$ and $K_1 = \max_{g \in \hat{\mathcal{G}}} K_1^g$. Let $\mathcal{H} = \{Tree(g, \hat{F}, M) | g \in \hat{\mathcal{G}}\}$ for some constant $M > 1$. Then if exactly one $h^* \in \mathcal{H}$ is true, the transition dynamics for $C.\alpha$ and $a$ can be learned with KWIK bound $N(K_0 M + K_1(D+1)+1)+N-1$ by applying the learning procedure of Algorithm 1. The proof of Theorem 1 is in section 4 of the supplementary material and is analogous to the KWIK bound proof for Propositional OO-MDPs [4].

Note that $DOORMAX_D$, being an $R_{max}$ based algorithm, requires two procedures - one to learn the transition dynamics from observed data and one to make predictions for planning. In this paper we present only the learning procedure (see Algorithm 1), leaving out the prediction procedure as it is analogous to that introduced for Propositional OO-MDPs [4]. For prediction, the essence of the procedure is that for an attribute $C.\alpha$ and action $a$ the input is a deictic object $o$ that is an instance of $C$ and a state $s$, and it is required that for each effect type in $\hat{\mathcal{G}}$ we can make a prediction that is not $\perp$ and further that all the predictions of $o.\alpha'$ are the same, otherwise return $\perp$. The full procedure is included in section 3 of the supplementary material.

---

**Algorithm 1:** $DOORMAX_D$: learning procedure for $C.\alpha$ and $a$.

**Input :** $o \in \mathcal{O}[C]$, $o.\alpha' \in Dom(C.\alpha)$, $s \in S$

1  pass $o$ and $s$ to the deictic predicates in $\hat{F}$ to retrieve a set of terms $T$
2  **foreach** $g \in \hat{\mathcal{G}}$ **do**
3      **foreach** $e \in g.E$ **do**
4          **if** $\mathcal{T}_e^g$ *does not exist* **then**
5              define $\mathcal{T}_e^g$ and initialise to the empty set, $\mathcal{T}_e^g \leftarrow \phi$
6          **end**
7          **if** $e(o.\alpha) = o.\alpha'$ **then**
8              **if** $\mathcal{T}_e^g = \phi$ **then**
9                  **if** $(\exists e' \in g.E$ *and* $T_{e'} \in \mathcal{T}_{e'}^g)$ *such that* $T_{e'} \subseteq T$ **then**
10                     remove $g$ from $\hat{\mathcal{G}}$
11                 **else**
12                     add $T$ to $\mathcal{T}_e^g$
13                 **end**
14             **else**
15                 **if** $g.\Pi(e) = 0$ **then**
16                     add $T$ to $\mathcal{T}_e^g$
17                 **else**
18                     $T_{temp} \leftarrow T$
19                     $T \leftarrow$ the only element in $\mathcal{T}_e^g$
20                     remove from $T$ any terms that mismatch terms in $T_{temp}$
21                     $\mathcal{T}_e^g \leftarrow \phi$
22                     add $T$ to $\mathcal{T}_e^g$
23                 **end**
24                 **if** $(\exists(e' \in (g.E - \{e\})$ *and* $T_{e'} \in \mathcal{T}_{e'}^g)$ *such that* $(T \subseteq T_{e'}$ *or* $T_{e'} \subseteq T))$ *or* $(|\mathcal{T}_e^g| > M)$ **then**
25                     remove $g$ from $\hat{\mathcal{G}}$
26                 **end**
27              **end**
28         **end**
29     **end**
30 **end**

---

The key insight behind Algorithm 1 is that for all $g \in \hat{\mathcal{G}}$ the set $\{\mathcal{T}_e^g\}_{e \in g.E}$ must at all times induce a binary tree subject to the constraints of the partition function $g.\Pi$. Each time we observe a set of terms $T$ and an associated effect $e$ we update $\mathcal{T}_e^g$ and in doing so may discover that the resulting set $\{\mathcal{T}_e^g\}_{e \in g.E}$ can no longer induce an appropriate binary tree at which point we remove $g$ from $\hat{\mathcal{G}}$. To provide some intuition on the operations in the algorithm consider a case where $F = \{f_1, f_2, f_3\}$ and there are two effects $E = \{e_1, e_2\}$ where $e_1$ is conjunctive and $e_2$ is disjunctive. Consider the following examples: Figure 4a: we currently have $\mathcal{T}_{e_1} = \{T_{e_1} = \{(f_1, 1), (f_2, 1), (f_3, 1)\}\}$ and $\mathcal{T}_{e_2} = \phi$. Suppose we then observe $e_2$ with $T = \{(f_1, 1), (f_2, 1), (f_3, 1)\}$. Then $\mathcal{T}_{e_2}$ is empty and $T_{e_1} \subseteq T$. Figure 4b: we currently have $\mathcal{T}_{e_1} = \{T_{e_1} = \{(f_1, 1), (f_2, 1), (f_3, 1)\}\}$ and $\mathcal{T}_{e_2} = \{T_{e_2} = \{(f_1, 1), (f_2, 1), (f_3, 0)\}\}$. Suppose we then observe $e_1$ with $T = \{(f_1, 1), (f_2, 0), (f_3, 0)\}$. As $e_1$ is conjunctive and $\mathcal{T}_{e_1}$ is not empty we first remove mismatching terms. Then $\mathcal{T}_{e_1} = \{T = \{(f_1, 1)\}\}$ and now $T \subseteq T_{e_2}$. Figure 4c: we currently have $\mathcal{T}_{e_1} = \{T_{e_1} = \{(f_1, 1)\}\}$ and $\mathcal{T}_{e_2} = \{T_{e_2} = \{(f_1, 0), (f_2, 0), (f_3, 0)\}\}$. Suppose we then observe $e_2$ with $T = \{(f_1, 1), (f_2, 0), (f_3, 1)\}$. We add

$T$ to $\mathcal{T}_{e_2}$ and now $T_{e_1} \subseteq T$. In all the above cases, we conclude that the effect type is invalid since the observed data can no longer induce a binary tree subject to the specified constraints. Note that we do not place any restrictions on the order in which the preconditions may appear in the tree, but there is no reordering that can recover an appropriate binary tree given the data.

## 4  Experiments

### 4.1  All-passenger any-destination Taxi domain

We conduct two sets of experiments on this domain. In the first set we have one destination and we fix the number of passengers, $n$. We generate a grounded MDP with an initial state by randomly sampling $n$ passenger locations and one destination location from one of six pre-specified locations and we also sample a random taxi start location together with one of four wall configurations as shown in Figure 1a. We apply 20 independent runs of the following procedure: we sample 10 test MDPs with random initial states. We then randomly sample a training MDP and run $DOORMAX_D$ on it for one episode until we reach the terminal state. Upon termination, we test performance by running $DOORMAX_D$ for one episode on each of the 10 test MDPs, stopping an episode early if we exceed 500 steps. We repeat this for 100 training MDPs. Since all the MDPs come from the same schema we can share transition dynamics between our MDPs - but we only update the transition dynamics on training MDPs. In our experiments we start with $n = 1$ passenger and increase to $n = 4$ passengers. We run our experiments for Propositional OO-MDPs and two versions of Deictic OO-MDPs. In the first, without transfer, we relearn the transition dynamics for each $n$ while for the second, with transfer, we transfer the previously learned transition dynamics each time we increase $n$. We report results in Figure 5 that averages over the 20 independent runs the average number of steps for the 10 test MDPs with error bars included.

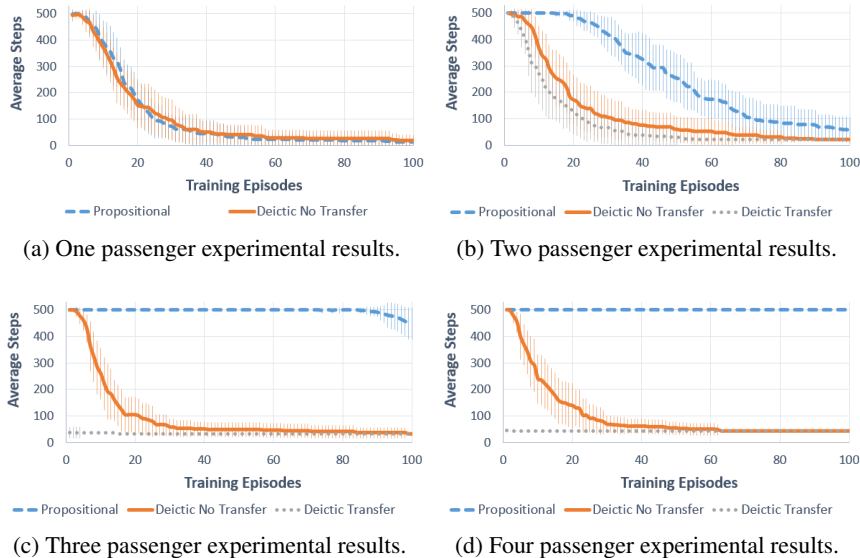

(a) One passenger experimental results.

(b) Two passenger experimental results.

(c) Three passenger experimental results.

(d) Four passenger experimental results.

Figure 5: Experimental results for the all-passenger any-destination Taxi domain with different number of passengers.

We see from the results that for this domain deictic representations outperform the propositional representation as we increase the number of passengers. This is because with the propositional representation we need to add more preconditions as we increase the number of passengers - in fact the propositional representation is unable to learn the task when $n = 4$ even after 100 training episodes. Furthermore, as the MDPs belong to the same schema it is beneficial to transfer the previous transition dynamics under the deictic representation. Specifically, once we get to $n = 4$ passengers the transition dynamics we transfer from the $n = 3$ experiment have learned the schema transition dynamics completely and we have zero-shot transfer of the transition dynamics.

We observe from Figure 5 that using the deictic representation without transfer is actually learning slightly faster as we add more passengers. This is somewhat misleading. What is actually happening is that as the tasks become more complex the agent is able to learn more about the transition dynamics over a single training episode, but that episode will require many more steps to complete. To illustrate this and also highlight the robustness of the deictic representation with transfer methodology we conduct a second set of experiments. These experiments are similar to those conducted before but we now use a larger $10 \times 10$ gridworld with five passengers and three destinations as in Figure 1b. In these experiments we stop after 100 steps for each episode of the training MDPs. Furthermore, for the deictic representation with transfer we simply transfer the learned transition dynamics of $n = 4$ passengers and do no additional learning on the new larger gridworld.

In Figure 6 we plot for all the experiments run the average number of steps relative to optimal number of steps. We see that the deictic representation with transfer is able to solve the larger gridworld optimally with no additional learning of the transition dynamics. Meanwhile the deictic representation with no transfer which was decreasing up to $n = 4$ now has a jump between $n = 4$ and $n = 5$ because the agent does not have the benefit of learning for a full training episode. We cap the graph's $y$ axis at 10 to make it more readable, but remark that the propositional representation exhibits exponentially worse performance relative to optimal as the tasks become more complex. We include additional details on the transition dynamics for this domain in section 1 of the supplementary material.

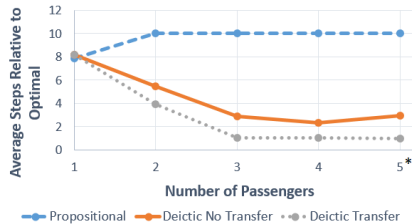

Figure 6: Average number of steps relative to optimal number of steps as we add more passengers - for $n = 5$ we also increase the gridworld size and add more destinations hence it is marked with a $*$.

## 4.2 Sokoban domain

To demonstrate the benefits of Deictic OO-MDPs, we conduct an experiment on a more challenging Sokoban domain. In this experiment we first learn the transition dynamics on an MDP with initial state as in Figure 2a and continue learning until we have a prediction for every state and action in the MDP. As it turns out, this simple toy MDP with $7,961$ states is enough to completely learn the schema transition dynamics of this domain under the Deictic OO-MDP representation. Once learned we zero-shot transfer the transition dynamics to a more complex Sokoban task as shown in 2c. This task comes from the "Micro-Cosmos" level pack and has approximately $10^6$ states while the optimal number of steps to solve this task 209. With no additional learning we run value iteration and solve for the optimal policy. Note that the ability to transfer here is critical. The larger MDP has approximately 125 times more states than the toy MDP. Running $R_{max}$ based algorithms directly on the larger MDP is very slow because at each step it is required to compute a policy with a planning algorithm such as value iteration that requires multiple iterations over the state-space to converge. By transferring the transition dynamics learned in the toy MDP we can solve the larger MDP with only a *single run* of value iteration which is extremely efficient. We include additional details on the transition dynamics for this domain in section 2 of the supplementary material.

## 5 Concluding remarks

In this paper we have introduced a novel deictic object-oriented representation for RL. We have shown that this representation can be described in terms of a schema that allows reuse of transition dynamics across grounded MDPs instantiated from that schema and that this allows for zero-shot transfer across such MDPs. Theoretically we have proved that under certain assumptions we can efficiently learn deterministic transition dynamics. We conducted experiments on an extension of the Taxi domain as well as a more challenging Sokoban domain and have shown that zero-shot transfer of transition dynamics is possible with our representation.

**Acknowledgments**

The authors with to thank Google Travel and Conference Grants for their support. The authors also wish to thank the anonymous reviewers for their thorough feedback and helpful comments.

## Footnotes

[1]In this paper we refer to propositions as logic statements over object classes, while we refer to first-order predicates as logic statements that depend on at least one grounded object.

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
