[Supplementary Material]

# Supplementary Material: Zero-Shot Transfer with Deictic Object-Oriented Representation in Reinforcement Learning

**Ofir Marom[1], Benjamin Rosman [1,2]**
[1]University of the Witwatersrand, Johannesburg, South Africa
[2]Council for Scientific and Industrial Research, Pretoria, South Africa

## 1  All-passenger any-destination Taxi domain preconditions and effects

In the main paper we introduce the all-passenger any-destination Taxi domain. Here we show for each attribute and action the relevant preconditions and effects.

We define the following preconditions:

- $WallNorthOfMe(taxi, s)$: returns 1 if there is a wall one square north of $taxi$ ($f_1$)
- $WallEastOfMe(taxi, s)$: returns 1 if there is a wall one square east of $taxi$ ($f_2$)
- $WallSouthOfMe(taxi, s)$: returns 1 if there is a wall one square south of $taxi$ ($f_3$)
- $WallWestOfMe(taxi, s)$: returns 1 if there is a wall one square west of $taxi$ ($f_4$)
- $TaxiOnMe(passenger, s)$: returns 1 if for a taxi is on the same square as $passenger$ ($f_5$)
- $MeAtDestination(passenger, s)$: returns 1 if $passenger.at\text{-}destination$ is true ($f_6$)
- $AnyPassengerInTaxi(s)$: returns 1 if a passenger's $in\text{-}taxi$ attribute is true ($f_7$)
- $MeInTaxi(passenger, s)$: returns 1 if $passenger.in\text{-}taxi$ is true ($f_8$)
- $TaxiOnAnyDestination(s)$: returns 1 if a taxi is on the same square as a destination ($f_9$)

Note that some preconditions above are actually propositions, not deictic predicates (for example $f_7$) since they do not depend on any grounded object. This does not contradict the Deictic OO-MDP framework since a deictic predicate may simply not refer to a grounded object in which case it is a proposition.

Define the following effects:

- $Rel_i(x) \rightarrow x + i$ for $i \in \{-1, 0, 1\}$ where $x$ is an Integer
- $Flip_i(x) \rightarrow (i = 0 : x), (else : !x)$ for $i \in \{0, 1\}$ where $x$ is Boolean

Then Table 1 shows for each attribute and action the relevant preconditions and effects

In the main paper we run experiments on the all-passenger any-destination Taxi domain. To test the performance and compare to Propositional OO-MDPs we need to choose a set of hypothesized preconditions. In constructing our experiments, we set out to mimic as closely as possible the representation described for the classical Taxi domain under Propositional OO-MDPs. Specifically, for the classic Taxi domain, the propositional representation depends for each attribute and action on the following propositions:

- $TouchNorth(Taxi, Wall)$: a taxi has a wall one square to its north ($p_1$)
- $TouchEast(Taxi, Wall)$: a taxi has a wall one square to its east ($p_2$)

Table 1: All-passenger any-destination Taxi domain preconditions and effects for each attribute and action.

| Action | Attribute | Preconditions | Effects |
|--------|-----------|---------------|---------|
| $North$ | $Taxi.y$ | $WallNorthOfMe$ | $Rel$ |
| $East$ | $Taxi.x$ | $WallEastOfMe$ | $Rel$ |
| $South$ | $Taxi.y$ | $WallSouthOfMe$ | $Rel$ |
| $West$ | $Taxi.x$ | $WallWestOfMe$ | $Rel$ |
| $Pickup$ | $Passenger.in\text{-}taxi$ | $TaxiOnMe, MeAtDestination,$ $AnyPassengerInTaxi$ | $Flip$ |
| $Dropoff$ | $Passenger.in\text{-}taxi$ | $MeInTaxi, TaxiOnAnyDestination$ | $Flip$ |
| $Dropoff$ | $Passenger.at\text{-}destination$ | $MeInTaxi, TaxiOnAnyDestination$ | $Flip$ |

- $TouchSouth(Taxi, Wall)$: a taxi has a wall one square to its south ($p_3$)
- $TouchWest(Taxi, Wall)$: a taxi has a wall one square to its west ($p_4$)
- $On(Taxi, Passenger)$: a taxi is on the same square as a passenger ($p_5$)
- $On(Taxi, Destination)$: a taxi is on the same square as a destination ($p_6$)
- $InTaxi(Passenger)$: a passenger has its in-taxi attribute set to true ($p_7$)

In order to remain comparable with Propositional OO-MDPs, our approach is to use deictic predicates when we can and use propositions when we must. For example, consider the $Passenger.in\text{-}taxi$ attribute. It is not possible to convert $TouchNorth(Taxi, Wall)$ to a deictic predicate since when predicting the $Passenger.in\text{-}taxi$ attribute we only have access to a grounded passenger object, and we don't have access to grounded wall or taxi objects. So in this case we use propositions. However we can convert $On(Taxi, Passenger)$ to a deictic predicate of the form $TaxiOnMe(passenger, s)$ which was defined in previously. We note that for the one passenger Taxi domain we have similar performance between Propositional OO-MDPs and Deictic OO-MPDs as shown in figure 5a in the main paper indicating that we have not biased Deictic-OO MDPs in our experiments.

As discussed in the main paper, when running the $DOORMAX_D$ learning algorithm we take advantage of conjunctive effects to invalidate multiple preconditions with a single observation. To illustrate this process we show in Tables 2 and 3 for the $Taxi.x$ attribute, action $East$ and effect $Rel_1(x)$ as well as for the $Passenger.in\text{-}taxi$ attribute, action $PICKUP$ and effect $Flip_1(x)$ how the algorithm converges to the true set of preconditions from a simulation of $DOORMAX_D$. Note for example in Table 2 we were able to learn the correct preconditions in only 7 observations whereas learning by memorisation would require $2^6$ separate observations.

## 2 Sokoban domain preconditions and effects

We show for each attribute and action the relevant preconditions and effects for the Sokoban domain. Note that in Sokoban it is easy for a person to get stuck in state from which the task is no longer solvable. To overcome this problem, we add a $Reset$ action that activates a person's $reset$ attribute to true and that immediately terminates the episode. For this domain, we set the reward dynamics to $-1$ for each step and 300 for getting all the boxes to a storage location. The high reward for completing the task is necessary to prevent the agent from learning an optimal policy whereby it just applies the $Reset$ action at the start of every episode.

We define the following preconditions:

- $WallNorthOfMe_i(person, s)$: returns 1 if there is a wall $i$ squares north of $person$
- $WallEastOfMe_i(person, s)$: returns 1 if there is a wall $i$ squares east of $person$

Table 2: All-passenger any-destination Taxi domain learning preconditions for $Taxi.x$ attribute, action $East$ and effect $Rel_1(x)$.

| obs | $f_1$ | $f_2$ | $f_3$ | $f_4$ | $p_5$ | $p_6$ | $p_7$ |
|-----|-------|-------|-------|-------|-------|-------|-------|
| 1 | 0 | 0 | 0 | 1 | 0 | 0 | 0 |
| 2 | - | 0 | 0 | 1 | 0 | - | 0 |
| 3 | - | 0 | 0 | - | 0 | - | 0 |
| 4 | - | 0 | 0 | - | 0 | - | - |
| 5 | - | 0 | 0 | - | 0 | - | - |
| 6 | - | 0 | 0 | - | 0 | - | - |
| 7 | - | 0 | - | - | - | - | - |

Table 3: All-passenger any-destination Taxi domain learning preconditions for $Passenger.in\text{-}taxi$ attribute, action $PICKUP$ and effect $Flip_1(x)$.

| obs | $p_1$ | $p_2$ | $p_3$ | $p_4$ | $f_5$ | $f_6$ | $p_7$ |
|-----|-------|-------|-------|-------|-------|-------|-------|
| 1 | 0 | 1 | 1 | 0 | 1 | 0 | 0 |
| 2 | - | 1 | - | 0 | 1 | 0 | 0 |
| 3 | - | - | - | - | 1 | 0 | 0 |

- $WallSouthOfMe_i(person, s)$: returns 1 if there is a wall $i$ squares south of $person$
- $WallWestOfMe_i(person, s)$: returns 1 if there is a wall $i$ squares west of $person$
- $BoxNorthOfMe_i(person, s)$: returns 1 if there is a box $i$ squares north of $person$
- $BoxEastOfMe_i(person, s)$: returns 1 if there is a box $i$ squares east of $person$
- $BoxSouthOfMe_i(person, s)$: returns 1 if there is a box $i$ squares south of $person$
- $BoxWestOfMe_i(person, s)$: returns 1 if there is a box $i$ squares west of $person$
- $PersonNorthOfMe(box, s)$: returns 1 if there is a person one square north of $box$
- $PersonEastOfMe(box, s)$: returns 1 if there is a person one square east of $box$
- $PersonSouthOfMe(box, s)$: returns 1 if there is a person one square south of $box$
- $PersonWestOfMe(box, s)$: returns 1 if there is a person one square west of $box$
- $BoxNorthOfMe(box, s)$: returns 1 if there is a box one square north of $box$
- $BoxEastOfMe(box, s)$: returns 1 if there is a box one square east of $box$
- $BoxSouthOfMe(box, s)$: returns 1 if there is a box one square south of $box$
- $BoxWestOfMe(box, s)$: returns 1 if there is a box one square west of $box$
- $ResetActivated(person, s)$: returns 1 if $person.reset$ is true

Define the following effects:

- $Rel_i(x) \rightarrow x + i$ for $i \in \{-1, 0, 1\}$ where $x$ is an Integer
- $Flip_i(x) \rightarrow (i = 0 : x), (else : !x)$ for $i \in \{0, 1\}$ where $x$ is Boolean

Then Table 4 shows for each attribute and action the relevant preconditions and effects

Table 4: Sokoban domain preconditions and effects for each attribute and action.

| Action | Attribute | Preconditions | Effects |
|--------|-----------|---------------|---------|
| *North* | *Person.y* | $WallNorthOfMe_1, WallNorthOfMe_2,$ $BoxNorthOfMe_1, BoxNorthOfMe_2$ | *Rel* |
| *East* | *Person.x* | $WallEastOfMe_1, WallEastOfMe_2,$ $BoxEastOfMe_1, BoxEastOfMe_2$ | *Rel* |
| *South* | *Person.y* | $WallSouthOfMe_1, WallSouthOfMe_2,$ $BoxSouthOfMe_1, BoxSouthOfMe_2$ | *Rel* |
| *West* | *Person.x* | $WallWestOfMe_1, WallWestOfMe_2,$ $BoxWestOfMe_1, BoxWestOfMe_2$ | *Rel* |
| *North* | *Box.y* | $PersonSouthOfMe, BoxNorthOfMe,$ $WallNorthOfMe$ | *Rel* |
| *East* | *Box.x* | $PersonWestOfMe, BoxEastOfMe,$ $WallEastOfMe$ | *Rel* |
| *South* | *Box.y* | $PersonNorthOfMe, BoxSouthOfMe,$ $WallSouthOfMe$ | *Rel* |
| *West* | *Box.x* | $PersonEastOfMe, BoxWestOfMe,$ $WallWestOfMe$ | *Rel* |
| *Reset* | *Person.reset* | *ResetActivated* | *Flip* |

# 3 Prediction with $DOORMAX_D$

In the main paper we leave the prediction procedure out due to space limitations. We include it here for completeness and note that it is analogous to that introduced under Propositional OO-MDPs. See Algorithm 2.

# 4 Proof of theorem

In the main paper we exclude the proof of Theorem 1 due to space limitations. We include it here for completeness and note that it is analogous to the KWIK bound proof under Propositional OO-MDPs.

Consider an effect type $g \in \hat{\mathcal{G}}$. If this is the correct effect type then it can be learned with KWIK bounds $K_0^g M + K_1^g (D + 1)$. This is because the $K_1^g$ conjunctive effects require at most $D + 1$ observations each to learn the terms they depend on while the terms for the $K_0^g$ disjunctive effects can be memorised $M$ times each. If we then consider all $N$ effect types in $\hat{\mathcal{G}}$, an upper bound on the number of observations required so that all effect types are either removed or return some prediction is $N(K_0 M + K_1(D + 1) + 1)$.

Now when we call the prediction procedure (see Algorithm 2) if two effect types provide a prediction that is not the same we remove one of them on the subsequent run of the learning procedure (see Algorithm 1 of the main paper) and this can occur at most $N - 1$ times. This gives a total KWIK bound of $N(K_0 M + K_1(D + 1) + 1) + N - 1$.

**Algorithm 2:** $DOORMAX_D$: prediction procedure for $C.\alpha$ and $a$.

**Input** : $o \in \mathcal{O}[C]$, $s \in S$
**Output** : $o.\alpha' \in Dom(C.\alpha)$ or $\perp$

1   let $V$ be a set that may contain elements in $Dom(C.\alpha)$ and set $V \leftarrow \phi$

2   pass $o$ and $s$ to the deictic predicates in $\hat{F}$ to retrieve a set of terms $T$

3   **foreach** $g \in \hat{\mathcal{G}}$ **do**
4     $pred \leftarrow \perp$
5     **foreach** $e \in g.E$ **do**
6       **if** $\exists T_e \in \mathcal{T}_e^g$ *such that* $T_e \subseteq T$ **then**
7         $pred \leftarrow e(o.\alpha)$
8         exit loop
9       **end**
10     **end**
11     **if** $pred = \perp$ **then**
12       return($\perp$)
13     **else**
14       add $pred$ to $V$
15       **if** $|V| > 1$ **then**
16         return ($\perp$)
17       **end**
18     **end**
19     return(the only element in $V$)
20   **end**

## 5   Source Code

A C# implementation of the Deictic OO-MDP framework for the Taxi domain experiments described in the main paper can be found here: https://github.com/OfirMarom/DeicticOOMDPs