[Reviews · NeurIPS 2018]

Reviewer 1



Post rebuttal: I now understand the middle ground this paper is positioned, and the difference to propositional OO representations where you don't necessarily care which instance of an object type you're dealing with, which significantly reduces the dimensionality of learning transition dynamics. But this is still similar to other work on graph neural networks for model learning in fully relational representations, like Relation Networks by Santoro et al., and Interaction Networks by Battaglia et al. which in worst case learn T * n * (n-1) relations for n objects for T types of relations. However, this paper does do a nice job of formalizing from the OO-MDP and Propositional MDP setting as opposed to the two papers I mentioned which do not, and focus on the physical dynamics case. I am willing to increase my score based on this, but still do not think it is novel enough to be accepted. --------------------------------------------------------------------------------------- This work proposes a deictic object-oriented representation where objects are grounded by their relation to a target object rather than global information. This is very similar to relational MDPs, but they learn transition dynamics in this relational attribute space rather than real state space. Compared to OO-MDPs this can zero-shot transfer to environments with more objects as long as the types of the objects stay constant. They are also able to give a bound on memorization of the transition dynamics in attribute space by modifying the DOORMAX algorithm to their deictic version. My main issue with this work is that I'm not sure how this differs from relational MDPs. The method learns dynamics of an environment in this relational attribute space which is given and borrows the DOORMAX algorithm from OO-MDPs, and are able to show nice results on two environments, but I'm not sure it is novel. If the authors could clearly explain the differences between relational MDPs and deictic OO-MDPs, it would be very helpful.

Reviewer 2



Summary of paper This paper introduces Deictic Object-Oriented MDPs, which define MDP dynamics in terms of effects that are relative to a particular "central" object (i.e. deictically defined). A variant of DOORMAX is introduced for the efficient learning of DOOMDPs and their use to guide exploration for sample-efficient MBRL. Finally, experiments demonstrate that the deictic and object-oriented nature of the models can facilitate effective transfer between tasks that share the same dynamics schema. Technical Quality As far as I am able to tell the paper is technically sound. DOOMDPs and DOORMAX_D are well-defined and well-motivated. The experiments effectively support the claims that deictic predicates permit the compact representation (and correspondingly efficient learning) of domains with varying numbers of objects. Clarity I found the paper to be well-written. The OOMDP framework is highly technical and notation-heavy, and I appreciated the authors' efforts to aid in the reader's comprehension with concrete examples. One place I noticed where this might be improved was in Section 2.2. I found it very difficult to take in the entirety of the framework so quickly and was wishing for a running example to make components concrete. I wonder if the authors would consider interleaving the description of the Taxi domain with the definition of DOOMDPs? I don't think it would require a big change in space usage, but might help the reader make sense of the framework as it is described rather than having to circle back later and map the examples to the general description. Originality The paper is fully open about the fact that these findings build heavily upon existing theory and algorithms (particularly POOMDPs and DOORMAX). There is a novel contribution, though. First, conceptually, the paper threads a needle between a purely propositional system, which of course suffers when the number of objects can change, and a fully first-order system, which would negatively impact learnability. The observation that deictic predicates can yield some of the benefits of a first-order framework without incurring its costs is not entirely novel in the history of logic-based systems of course, but it is, as far as I know, novel in the context of OOMDPs. Second, the paper offers a particular instantiation of this idea, working through the details of how deictic predicates would be defined and learned. Significance I would say that the practical significance of this paper is rather low. Though appealing in many ways, OOMDPs are not to the point where they can be applied to problems of genuine interest -- they require far too much prior domain knowledge and do not fail gracefully when their biases are not reflected in the environment. Expanding the reach of OOMDPs from small, idealized toy problems to bigger, idealized toy problems is not going to change current practice. However, I do think there is something to be said about exploring the edges of what a model with strong generalization can achieve. Rmax showed that sample efficient RL was possible, and this work continues a thread extending that analysis to richer and richer state spaces, where the structure is exploited to avoid dependence on the size of the state space, and now can be shown to facilitate generalization from easy problems to much larger and complex problems as the deictic operators allow for the expansion of the number of objects. So from a broader view of MBRL, this paper plants a theoretical flag that says "if you could have a model like this, you could enjoy some massive benefits." To the extent to which the paper teaches us about what we could possibly hope to achieve with MBRL, and what model properties might be necessary to get there, I think it may be of interest to the community. Overall I found the paper to be well-written and engaging. It represents a non-trivial extension to existing work. Its contributions are largely conceptual and theoretical and not likely to immediately impact practice. Nevertheless, the findings do expand what is possible in the context of provably efficient MBRL and explore a type of model-structure that may be more generally valuable. ---After Author Reponse--- Having considered the other reviews and the author response, I still feel that this paper offers a novel and worthwhile contribution in formalizing the deictic setting as a middle ground between fully propositional and fully relational representations. I don't think this result would necessarily have a big impact practically (OOMDPs are hard to scale) or conceptually (I think it is already widely understood that representations like this are desirable) but I do think the results in this paper teach us something about the theoretical boundaries of efficient RL and that's worth having in the literature, so I still tend to vote for acceptance.

Reviewer 3



The paper proposes what it calls "Diectic Object-Oriented MDPS", in contrast to previously published Propositional OO-MDPS. The advantage of using the diectic approach is that the inference is relative to the grounded context. The method assumes full state knowledge as well as the semantic classes of each object, but learns the transition probabilities from data. The experiments show the method working on the taxi problem, with the learned transitions generalize accross different numbers of passengers. This is good, but not particularly groundbreaking. How would a regular model-based approach perform on this task? The sokoban result is impressive: a model trained on one size of grid transfers to a larger grid with now training. However, the paper does not show how well the VAlue iteration from the transferred model actually performs. This paper is not at all in my area, so I am not sure of some of the terminology in this work. I am also not familiar what constitutes a "good paper" in this area. --- After reading the other reviews and author response, I continue to find this work an interesting formalization of a real problem. I have trouble seeing how this formalization would be enacted in a real world scenario. I am in favor of acceptance, but I would prefer to see how the method fares in non-ideal scenarios (e.g. if some approximations need to be made, or there is some uncertainty about the world)